# Risk of Mitral Valve Prolapse in Patients with Keratoconus in Taiwan: A Population-Based Cohort Study

**DOI:** 10.3390/ijerph17176049

**Published:** 2020-08-20

**Authors:** Yuh-Shin Chang, Ming-Cheng Tai, Shih-Feng Weng, Jhi-Joung Wang, Sung-Huei Tseng, Ren-Long Jan

**Affiliations:** 1Department of Ophthalmology, Chi Mei Medical Center, Tainan 710, Taiwan; yuhshinchang@yahoo.com.tw (Y.-S.C.); shtseng1@gmail.com (S.-H.T.); 2Graduate Institute of Medical Sciences, College of Health Sciences, Chang Jung Christian University, Tainan 711, Taiwan; 3Department of Ophthalmology, Tri-Service General Hospital, National Defense Medical Center, Taipei 114, Taiwan; mingtai1966@yahoo.com.tw; 4Department of Healthcare Administration and Medical Informatics, Kaohsiung Medical University, Kaohsiung 807, Taiwan; also99@gmail.com; 5Department of Medical Research, Chi Mei Medical Center, Tainan 710, Taiwan; 400002@mail.chimei.org.tw; 6Department of Ophthalmology, National Cheng Kung University Hospital, College of Medicine, National Cheng Kung University, Tainan 701, Taiwan; 7Department of Pediatrics, Chi Mei Medical Center, Liouying, Tainan 736, Taiwan

**Keywords:** mitral valve prolapse, keratoconus, Taiwan Longitudinal Health Insurance Database

## Abstract

This retrospective, nationwide, matched-cohort study included 4488 new-onset keratoconus (KCN) patients, ≥12 years old, recruited between 2004 and 2011 from the Taiwan National Health Insurance Research Database. The control group included 26,928 non-KCN patients selected from the Taiwan Longitudinal Health Insurance Database 2000. Information for each patient was collected and tracked from the index date until December 2013. The incidence rate of mitral valve prolapse (MVP) was 1.77 times (95% confidence interval (CI) = 1.09–2.88; *p* = 0.0206) higher in KCN patients ≥40 years old and 1.49 times (95% CI = 1.12–1.98; *p* = 0.0060) higher in female KCN patients than in controls. After using the Cox proportional hazard regression analysis to adjust for potential confounders, including hypertension, hyperlipidemia, and congestive heart failure, KCN maintained an independent risk factor, MVP being 1.77 times (adjusted hazard ratio (HR) = 1.77, 95% CI = 1.09–2.88) and 1.48 times (adjusted HR = 1.48, 95% CI = 1.11–1.97) more likely to develop in patients ≥40 years old and female patients in the study cohort, respectively. We found that KCN patients ≥40 years of age and female KCN patients have increased risks of MVP. Therefore, it is recommended that KCN patients should be alerted to MVP.

## 1. Introduction

Keratoconus (KCN) is a bilateral, asymmetric, and progressive ectatic condition in which the affected cornea has a conical shape, leading to notable visual impairment [1]. The disease has significant visual morbidity and is the main reason for keratoplasty in the developed world [2]. KCN affects all ethnicities and both genders. It typically appears in adolescence and advances until the third or fourth decade. Despite a high prevalence, the etiology of KCN is not fully understood. The cellular etiology of the disease is evaluated genetically, biochemically, and physically, and it has been implicated that the disease may be multifactorial in origin [3]. KCN may occur sporadically or due to genetic predispositions. Environmental factors and stimuli associated with the inception and progression of KCN contribute to the disease. Hence, KCN is a multifactorial disorder and involves multiple genes and various mechanisms that lead to clinical disease etiology.

Mitral valve prolapse (MVP), a common heritable valvulopathy, is the most important cause of primary mitral regurgitation (MR) requiring surgery [4,5]. Primary connective tissue abnormalities of the leaflets, chordae tendineae, and annulus of the mitral valves affect approximately 2.4% of the population [6,7]. MVP is characterized by fibromyxomatous changes of the valve, structural alterations of collagen in the leaflet, and structurally abnormal chordae, which may be related to dysregulation of the extracellular matrix (ECM) components [8,9]. Most individuals with MVP have no MR or mild MR [10], but MVP can lead to severe MR, arrhythmias, congestive heart failure, infective endocarditis, and even sudden cardiac death [4,8,9].

Both KCN and MVP are noninflammatory conditions with unclear etiology or pathophysiology. Structures of the cornea stroma and human heart valves are rich in collagen components. The dysregulation of the collagen components plays a key role in the development of these diseases [8,9,11,12]. Apart from their common pathogenic mechanisms, KCN and MVP are linked to several systemic collagen diseases, including Marfan syndrome, Ehlers–Danlos syndrome, pseudoxanthoma elasticum, and osteogenesis imperfecta [3,13,14]. Hence, it is clinically related to explore whether KCN is a risk in the development of MVP.

Previous studies have discussed the association of MVP in KCN patients [15,16,17], but their results were limited by small sample sizes or the lack of comparative controls. Beardsley et al. reported an MVP prevalence of 38% in 32 KCN patients, which was statistically different from the 13% in their group of age- and sex-matched controls [17]. Sharif et al. screened 50 advanced keratoconus patients requiring corneal transplantation for MVP and determined an overall prevalence of 58%, which was significantly higher than the 7% reported in their age- and sex-matched controls [15]. Rabbanikhah et al. conducted a case–control study of 32 patients with KCN with corneal hydrops and their age- and sex-adjusted controls, and showed that patients with hydrops had an odds ratio of 26.7 for MVP (95% CI = 9.5–75.2) [16]. However, in the study by Street et al., the difference in the prevalence of mitral valve prolapse in 95 patients with keratoconus, compared with 96 controls, was not statistically significant [18]. These studies were limited by small sample sizes. Therefore, we designed this study for clarifying the association between KCN and MVP, using a nationwide population-based data set.

## 2. Methods

### 2.1. Database

In Taiwan, the single-payer National Health Insurance (NHI) scheme, started since 1 March 1995, renders all residents with comprehensive medical care. Since 2007, the program has enrolled 22.60 million individuals and comprises >98% of the total Taiwanese population. The National Health Research Institute (NHRI) provides the National Health Insurance Research Database (NHIRD), which is the source of our data. For each beneficiary, the NHIRD records code information regarding the patient’s residential area, gender, and birthdate, as well as the International Classification of Diseases, Ninth Revision, Clinical Modification (ICD-9-CM) codes, including diagnoses and details of prescriptions, surgical or noninvasive procedures, and expenses, irrespective of where the patient is during hospitalization or under ambulatory care. Hence, the NHIRD can clearly reveal the conditions of medical utilization for the 22.60 million residents, and it is one of the largest databases in the world. The Institutional Review Board of Chi Mei Medical Center granted this study exemption from ethics review on the basis that the personal data was unidentified, and therefore, informed consent was waived. We have followed the guidelines of the Helsinki Declaration.

### 2.2. Selection of Patients and Variables

We enrolled two groups in this retrospective cohort study: a new-onset KCN group and a matched non-KCN control group. Patients’ information from both groups was collected from 1 January 2004 to 31 December 2011, and both groups were traced until the end of 2013. Patients with unknown gender or missing data and individuals diagnosed with MVP (code 424.00) earlier than KCN were excluded. We chose KCN patients ≥12 years old because KCN typically appears in adolescence. In total, there were 4488 KCN patients ≥12 years old with the ICD-9-CM code 371.6 diagnosis.

For each KCN patient, one non-KCN control was randomly chosen from the longitudinal Health Insurance Database 2000, a subset of NHIRD that includes overall claim data for 1,000,000 beneficiaries from the 2000 registry, constructed from NHIRD by NHRI using a systematic sampling method. Controls (n = 26,928) were matched with KCN patients by age, sex, comorbidities such as hyperlipidemia, hypertension, and congestive heart failure, as well as index date. The index date of these patients was defined as the first day of diagnosis for KCN patients. If controls were diagnosed with KCN or MVP before the index date, they were excluded. To determine the MVP incidence, each participant in both groups was tracked from the index date until death or the end of 2013, whichever occurred first. Each participant’s demographic data were recorded. Additionally, data regarding their comorbidities, including hyperlipidemia (code 272), hypertension (codes 401–405), and congestive heart failure (code 428), were collected because these are decisive risks for MVP [6,7,19,20]. Comorbidities were considered to be included only if the condition occurred in an inpatient setting or if they appeared in three or more ambulatory care claims within one year earlier than the index date.

### 2.3. Statistical Analysis

We used SAS 9.4 for Windows (SAS Institute, Inc., Cary, NC, USA) for statistical analyses. The baseline demographical characteristics and relevant comorbidities were compared between the KCN patients and controls using Pearson’s chi-square test. The MVP incidence was calculated as all KCN patients discovered during the follow-up, divided by the entire person-years (PYs) for each group by age, gender, and selected comorbidities. The Poisson regression method was utilized to calculate the incidence rate ratio (IRR), which compared the MVP risk between KCN patients and non-KCN controls. The study used the Cox proportional hazard model to estimate the hazard ratios (HRs), with 95% confidence intervals (CIs) for the association between KCN and MVP after controlling for relevant variables. The cumulative incidence rates for MVP were calculated using Kaplan–Meier analyses, and differences in the cumulative incidence rate curves were analyzed using log-rank tests. Statistical significance in this study was defined as *p* < 0.05.

## 3. Results

### 3.1. Demographic Data

From the start of 2004 to the end of 2011, after omitting improper subjects, 4488 KCN patients and 26,928 controls were enrolled. Table 1 reveals the baseline demographical characteristics and comorbidities of the KCN and non-KCN (control) groups. The average age of the KCN patients and controls was 27.38 (standard deviation (SD), 13.08) and 27.50 (SD, 12.88) years, respectively. Of the 4488 KCN patients, 2468 (54.99%) were men and 2020 (45.01%) were women, with 1250 (27.85%) aged 12–19 years, 1830 (40.78%) 20–29 years, 873 (19.45%) 30–39 years, and 535 (11.92%) ≥40 years. With regard to the comorbidities, 145 (3.23%) KCN patients had hypertension, 66 (1.47%) KCN patients had hyperlipidemia, and 7 (0.16%) KCN patients had congestive heart failure.

### 3.2. Incidence Rates for MVP

During the follow-up period, there was a higher MVP incidence rate in patients with KCN aged ≥40 years (73.27/10,000 PYs) than in the age-matched controls (41.31/10,000 PYs). Additionally, a significant difference was present in the MVP incidence rate (IRR 1.77, 95% CI = 1.09–2.88, *p* = 0.0206; Table 2) between the two groups. The order of MVP incidence rates in other KCN age groups is as follows: aged 20–29 years (34.77/10,000 PYs), 30–39 years (30.74/10,000 PYs), and 12–19 years (24.69/10000 PYs). However, there were no significant differences in MVP incidence rates between these aged groups and their controls.

The MVP incidence rate was 54.29/10,000 PYs for women with KCN and 36.47/10,000 PYs for female controls, leading to a significant difference between women with KCN and female controls (IRR = 1.49, 95% CI = 1.12–1.98, *p* = 0.0060). IRR for MVP in men with KCN indicated that the risk of MVP was not significantly greater than that for the corresponding controls (Table 2).

In the KCN group, the MVP incidence rates decreased in the following order: patients with hypertension (25.58/10,000 PYs), hyperlipidemia (32.26/10,000 PYs), and congestive heart failure (0/10,000 PYs). IRR for MVP in KCN patients with these comorbidities indicated that the risk was not significantly greater than for the corresponding controls.

### 3.3. Cumulative Incidence Rates for MVP

Kaplan–Meier analyses showed higher cumulative incidence rates of MVP for patients aged ≥40 years in the KCN group than in the controls. The log-rank test findings were also statistically significant *(p* = 0.0192; Figure 1).

Higher cumulative incidence rates of MVP for females in the KCN group than the non-KCN group were shown by Kaplan–Meier analyses, and log-rank test findings were also significant (*p* < 0.0058; Figure 2).

### 3.4. Hazard Ratios for MVP

Table 3 demonstrates the crude and adjusted HRs for MVP in participants aged ≥40 years in both groups during the follow-up period. After adjustment for gender and the selected comorbidities, KCN was still an independent risk of MVP in participants aged ≥40 years (adjusted HR = 1.77, 95% CI = 1.09–2.88). Being female was a significant risk factor for MVP in participants aged ≥40 years in both groups (adjusted HR, 1.89, 95% CI = 1.20–2.97, *p* < 0.05). Hypertension, hyperlipidemia, and congestive heart failure were not independent risks for MVP in participants aged ≥40 years.

Table 4 shows the crude and adjusted HRs for MVP in female participants during the period of follow-up. After adjustments for age and the selected comorbidities, KCN stood as a risk for MVP in female participants (adjusted HR = 1.48, 95% CI = 1.11–1.97). Regarding age, the only significant risk factor for MVP in female participants of both groups was age ≥40 years (adjusted HR, 1.85, 95% CI, 1.26–2.71, *p* < 0.05). Hyperlipidemia, hypertension, and congestive heart failure were not independent risks for MVP.

## 4. Discussion

According to a thorough review of relevant research, our study is the largest-scale population-based study to explore the relationship between KCN and subsequent MVP. We analyzed 4488 KCN patients, ≥12 years old, and 26,928 matched controls. The study results indicated a significantly increased risk of MVP in KCN patients ≥40 years of age and female KCN patients compared with controls. The study also showed that KCN is still an independent risk factor for MVP in participants ≥40 years old (Table 3) or female participants (Table 4) in the total cohort after accounting for age or sex, hypertension, hyperlipidemia, and congestive heart failure.

The corneal stroma, the main structure that provides corneal refraction, corneal transparency, and mechanical properties, plays an essential role in corneal stability and shape. The structure of the corneal stroma is composed of collagen tissue; therefore, corneal stability may be affected by dysregulation of the collagen component. Like the cornea, the human heart valves are rich in collagen tissues, with a majority of types I and V collagen and a small proportion of type III [11,12]. It is worth noting that in myxomatous degeneration, pathological weakening of the connective tissue plays an important role in the pathophysiology of both KCN and MVP [8,21]. In addition, several reports have shown restricted lysyl oxidase distribution and reduced total lysyl oxidase activity in keratoconic corneas [22,23]. These pose a possible explanation of inadequate collagen crosslinking, which is a major pathophysiological consideration in KCN. Similarly, decreased lysyl oxidase activity and impaired lysyl oxidase are found in mitral leaflets of patients with MVP [24]. These similar changes in the extracellular matrices may imply an association between MVP and KCN. Finally, several studies have shown that the biomechanical characteristics of keratoconic corneas, including corneal hysteresis and corneal resistance factor, were less than those in normal corneas [13,25,26,27]. Kalkan Akcay et al. reported that corneal hysteresis and corneal resistance factor values in patients with MVP were significantly less than those in the control group [13]. Similar changes in the biomechanical characteristics of the cornea may reflect an association between KCN and MVP.

The MVP incidence rate was significantly higher in KCN patients ≥40 years old (Table 2). Moreover, KCN maintained a significant risk for MVP development after accounting for gender and comorbidities in participants ≥40 years old of both groups (Table 4). Although no previous study has shown the effect of age on the incidence of MVP in patients with KCN, there are several studies suggesting that MVP is an age-dependent disorder [28,29,30]. Nishimura et al. reported that MVP is a result of progressive myxomatous valve changes and annular dilation [29]. Wilcken reported that MVP development and progression to severe MR occurs after the age of 50 years [28]. In addition, Devereux et al. showed that MVP is an inherited autosomal dominant condition with age-dependent expressions [30]. We have attempted to explain that KCN patients ≥40 years old have a risk factor for MVP development because MVP is an age-dependent progressive disorder.

We have found that the MVP incidence rate is higher in females with KCN (Table 2). Our findings suggest that KCN is a significant risk factor in developing MVP after accounting for age and comorbidities in the female sex of both groups (Table 4). There has not been a previous study that has evaluated the influence of gender in the association between MVP and KCN. However, there are several studies that have demonstrated an association between MVP and women of childbearing age [31,32,33]. Yuan et al. reported that cardiac problems in pregnant women, especially MVP, are a topic of concern [32]. Nanna’s study showed that MVP was more common and played an important etiology of valvular heart disease in childbearing-aged women [31]. We have tried to explain the higher MVP incidence rate in female KCN patients as related to the more common MVP incidence in pregnant and childbearing-aged women.

There are several strengths to our study. First, the study was able to demonstrate superior statistical power and precision in risk appraisal because it was a nationwide, population-based study that included a large sample of KCN patients. Secondly, there was decreased selection bias in referral centers, which reduced the chances of misdiagnosis, because patients with visual disturbances visit ophthalmologists and patients with cardiovascular problems visit cardiologists. Thirdly, the study was a cohort study with longitudinal data of up to 10 years. Finally, we have attempted to limit potential confounding bias by controlling for hypertension, hyperlipidemia, and congested heart failure.

There are several limitations to this study. We cannot confirm if the controls had a history of KCN before January 1996, because the medical histories of each sampled patient can only be tracked back to the year 1996. Selection bias might also have been involved in our study. As the controls were selected from patients who did not have KCN during the entire follow-up period, the controls would be comparatively healthier than the “real” general population because they were not only free of KCN at matching in the beginning, but also during the entire study duration. Accordingly, the incidence rate of MVP would be lower in these controls than in the general population. Therefore, the selection of healthier controls could result in the overestimation of the association between the risks of KCN and MVP. Additionally, several important systemic connective diseases such as Marfan syndrome, Ehlers–Danlos syndrome, pseudoxanthoma elasticum, and osteogenesis imperfecta could not be assessed as confounding factors due to their rare incidences. Finally, the diagnosis of KCN, MVP, and other comorbidities relied on ICD-9-codes, and incorrect classification is a possibility. The results of our study may not apply to other populations because this is a Taiwanese study.

## 5. Conclusions

This study showed that the risk of MVP was significantly higher in KCN patients ≥40 years of age and female KCN patients than their controls. KCN remained an independent risk factor in patients aged ≥40 years and the female sex in the cohort after adjusting for other confounders. These results suggest that clinicians should inform KCN patients about MVP, especially KCN patients ≥40 years of age and female KCN patients. In addition, mitral valve prolapse screening recommendations for patients with keratoconus would be helpful to clinicians.

## Figures and Tables

**Figure 1 ijerph-17-06049-f001:**
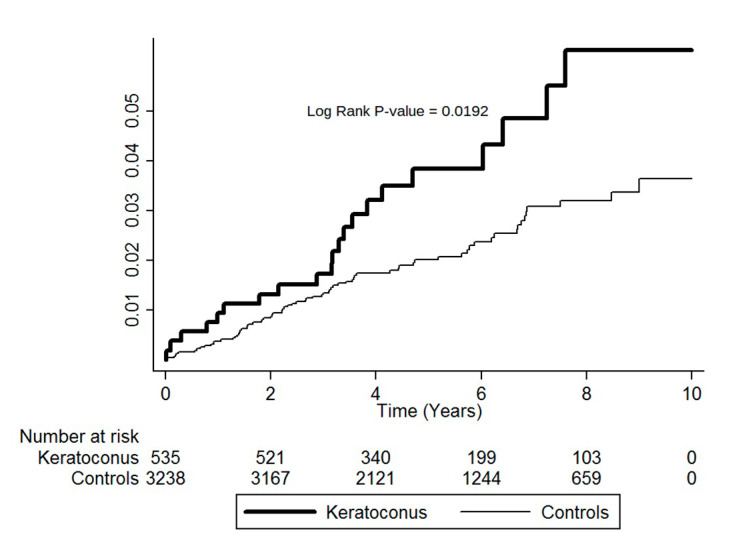
Cumulative incidence of mitral valve prolapse in keratoconus patients ≥40 years old and controls during the follow-up period.

**Figure 2 ijerph-17-06049-f002:**
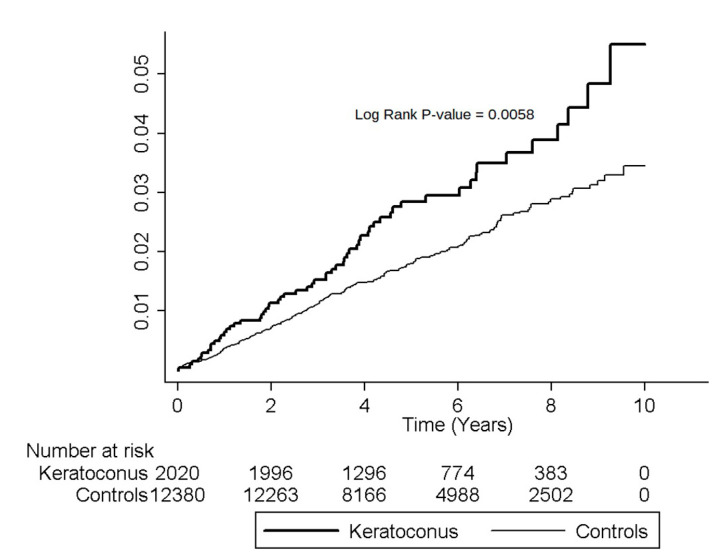
Cumulative incidence of mitral valve prolapse in female keratoconus patients and controls during the follow-up period.

**Table 1 ijerph-17-06049-t001:** Demographic characteristics and comorbid disorders between the keratoconus group and controls.

Characteristics	Keratoconus (N = 4488)	Controls (N = 26,928)	*p*-Value
	n (%)	n (%)	
Age	27.38 ± 13.08	27.50 ± 12.88	0.5513
Age group			
12–19 years	1250 (27.85)	7450 (27.67)	0.9892
20–29 years	1830 (40.78)	11,024 (40.94)	
30–39 years	873 (19.45)	5216 (19.37)	
≥40 years	535 (11.92)	3238 (12.02)	
Gender			
Male	2468 (54.99)	14,548 (54.03)	0.2294
Female	2020 (45.01)	12,380 (45.97)	
Baseline comorbidity			
Hypertension	145 (3.23)	871 (3.23)	0.9896
Hyperlipidemia	66 (1.47)	403 (1.50)	0.8942
Congestive heart failure	7 (0.16)	30 (0.10)	0.4203

Note: The demographic characteristics and comorbid disorders between the keratoconus and control groups were compared by Pearson chi-square tests.

**Table 2 ijerph-17-06049-t002:** Risk of mitral valve prolapse for the keratoconus group and the control group.

Characteristics	Keratoconus				Controls				IRR (95% CI)	*p*-Value
	N	MVP	PYs	Rate ^a^	N	MVP	PYs	Rate ^a^		
All	4488	88	24,658	35.69	26,928	433	149,769	28.91	1.23 (0.98–1.55)	0.0717
Age										
12–19 years	1250	17	6884	24.69	7450	115	42,431	27.10	0.91 (0.55–1.52)	0.7202
20–29 years	1830	36	10,353	34.77	11,024	174	62,462	27.86	1.25 (0.87–1.79)	0.2259
30–39 years	873	14	4555	30.74	5216	71	27,204	26.10	1.18 (0.66–2.09)	0.5760
≥40 years	535	21	2866	73.27	3238	73	17,672	41.31	1.77 (1.09–2.88)	0.0206
Gender										
Male	2468	29	13,790	21.03	14,548	185	81,775	22.62	0.93 (0.63–1.38)	0.7146
Female	2020	59	10,868	54.29	12,380	248	67,994	36.47	1.49 (1.12–1.98)	0.0060
Comorbidity										
Hypertension	145	2	782	25.58	871	27	4708	57.35	0.45 (0.11–1.88)	0.2705
Hyperlipidemia	66	1	310	32.26	403	5	2006	24.93	1.29 (0.15–11.07)	0.8146
Congestive heart failure	7	0	26	0	30	2	155	129.03	-	-

Note: A Poisson regression analysis was performed to calculate the incidence rate ratio. Abbreviations: MVP, mitral valve prolapse; IRR, incidence rate ratio; PYs: person-years; ^a^ rate per 10,000 person-years.

**Table 3 ijerph-17-06049-t003:** Crude and adjusted hazard ratios of Cox proportional hazard regressions and 95% confidence intervals for mitral valve prolapse at age ≥40 years during the follow-up period.

Cohort	Crude Hazard Ratio (95% CI)	Adjusted Hazard Ratio (95% CI)
Keratoconus		
Yes	1.77 * (1.09–2.88)	1.77 * (1.09–2.88)
No	1.00	1.00
Gender		
Male	1.00	1.00
Female	1.88 * (1.20–2.96)	1.89 * (1.20–2.97)
Comorbidity		
Hypertension		
Yes	1.15 (0.72–1.83)	1.24 (0.76–2.03)
No	1.00	1.00
Hyperlipidemia		
Yes	0.64 (0.26–1.58)	0.58 (0.23–1.47)
No	1.00	1.00
Congestive heart failure		
Yes	1.34 (0.19–9.58)	1.26 (0.17–9.15)
No	1.00	1.00

Note: The adjusted hazard ratio for developing mitral valve prolapse at age ≥40 years was calculated using the Cox proportional hazard regression analysis. Abbreviations: CI, confidence interval; * *p* < 0.05

**Table 4 ijerph-17-06049-t004:** Crude and adjusted hazard ratios of Cox proportional hazard regressions and 95% confidence intervals for mitral valve prolapse in females during the follow-up period.

Cohort	Crude Hazard Ratio (95% CI)	Adjusted Hazard Ratio (95% CI)
Keratoconus		
Yes	1.49 * (1.12–1.98)	1.48 * (1.11–1.97)
No	1.00	1.00
Age		
12–19 years	1.00	1.00
20–29 years	1.33 (0.96–1.83)	1.32 (0.96–1.81)
30–39 years	1.37 (0.96–1.94)	1.35 (0.95–1.93)
≥40 years	1.98 * (1.39–2.82)	1.85 * (1.26–2.71)
Comorbidity		
Hypertension		
Yes	1.79 * (1.14–2.82)	1.40 (0.82–2.30)
No	1.00	1.00
Hyperlipidemia		
Yes	0.91 (0.34–2.45)	0.55 (0.19–1.54)
No	1.00	1.00
Congestive heart failure		
Yes	5.73 * (1.43–23.02)	3.63 (0.88–15.00)
No	1.00	1.00

Note: The adjusted hazard ratio for developing mitral valve prolapse in females was calculated using the Cox proportional hazard regression analysis. Abbreviations: CI, confidence interval; * *p* < 0.05.

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
