# Peer review of "Risk of Mitral Valve Prolapse in Patients with Keratoconus in Taiwan: A Population-Based Cohort Study"

_ijerph, 2020, doi:10.3390/ijerph17176049_

Round 1

Reviewer 1 Report

Overall, well-written longitudinal study with a significantly sized sample to enhance validity of the results.

It would have useful to know the percentage of patients with keratoconus and mitral valve prolapse that developed mitral regurgitation requiring surgical intervention in comparison to the patients with mitral valve prolapse, but without keratoconus.

In addition, mitral valve prolapse screening recommendations for patients with keratoconus would be helpful for clinicians.

Author Response

My co-authors and I sincerely thank you for reviewing and commenting on our research article with Manuscript ID: ijerph-864158 

Reply: Thank you for your efforts and for taking valuable time to contribute to the reviewing process of our article. We have discussed and addressed your comments in detail to improve our manuscript.

Comments and Suggestions for Authors

Overall, well-written longitudinal study with a significantly sized sample to enhance validity of the results.

1. It would have useful to know the percentage of patients with keratoconus and mitral valve prolapse that developed mitral regurgitation requiring surgical intervention in comparison to the patients with mitral valve prolapse, but without keratoconus.

Reply: Thank you for the valuable comment. As stated in the table below, we found that 1 out of 88 KCN patients with MVP developed mitral regurgitation with surgical intervention, while 2 out of 433 non-KCN patients with MVP did. Due to the low number of patients who underwent MR with surgery, it is meaningless to compare the statistical significance. The low frequency may be attributed to the fact that we did not follow patients long enough and most patients with MVP have a good prognosis.

KCN

(n = 4488)

Non-KCN

(n = 26928)

N (%)

N (%)

MVP

88 (1.96%)

433 (1.60%)

Mitral regurgitation with surgical intervention

 1 (1/88, 1.13%)

2 (2/433, 0.46%)

2. In addition, mitral valve prolapse screening recommendations for patients with keratoconus would be helpful for clinicians.

Reply: Thank you for the valuable comment. We have added the sentence in accordance with your suggestion in the revised manuscript on page 10, lines 270−271.

Reviewer 2 Report

The authors have sought to clarify the association between keratoconus and mitral valve prolapse, using a nationwide population-based data in Taiwan.

The question being asked is valuable and the method was appropriate.

Introduciton

Line 64: "Previous studies have discussed the association of MVP in KCN patients ...", previous studies need to be describe more clearly about the main outcomes and limitations.

Method

Line 88: "as were individuals diagnosed ...", did it meant as well?

Line 89: Why chosed patients >12 years rather than other age?

Line 100: "data regarding their comorbidities including hyperlipidaemia ...",  more studies should be cited to support the comorbidities selected for controls to determine whther it is appropriate.

Author Response

My co-authors and I sincerely thank you for reviewing and commenting on our research article with Manuscript ID: ijerph-864158 entitled, Risk of Mitral Valve Prolapse in Patients with Keratoconus in Taiwan: A Population-Based Cohort Study.” We have revised the manuscript in accordance with your recommendations. The changes are marked in red text in the revised manuscript. The following are our replies to the reviewers’ comments. We hope that our revised article will be accepted for publication in International Journal of Environmental Research and Public Health.

Reviewer(s)' Comments to Author:

Reviewer: 2

Reply: Thank you for your efforts and for taking valuable time to contribute to the reviewing process of our article. We have discussed and addressed your comments in detail to improve our manuscript.

Comments and Suggestions for Authors

The authors have sought to clarify the association between keratoconus and mitral valve prolapse, using a nationwide population-based data in Taiwan.

The question being asked is valuable and the method was appropriate.

Introduciton

1. Line 64: "Previous studies have discussed the association of MVP in KCN patients ...", previous studies need to be describe more clearly about the main outcomes and limitations.

Reply: Thank you for pointing this out. We have moved the statements about the main outcomes and limitations of previous studies “Beardsley et al. reported a MVP prevalence of 38% in 32 KCN patients, which was statistically different from the 13% in their group of age- and sex-matched controls [17]. Sharif et al. screened 50 advanced keratoconus patients requiring corneal transplantation for MVP and determined an overall prevalence of 58%, which was significantly higher than the 7% reported in their age and sex-matched controls [15]. Rabbanikhah et al. conducted a case-control study of 32 patients with KCN with corneal hydrops and their age- and sex-adjusted controls, and showed that patients with hydrops had an odds ratio of 26.7 for MVP (95% CI = 9.5–75.2) [16]. However, in the study by Street et al., difference in the prevalence of mitral valve prolapse in 95 patients with keratoconus, compared with 96 controls was not statistically significant [18]. These studies were limited by small sample sizes.” from the Discussion section to the Introduction section in the revised manuscript on page 2, lines 65−75.

Method

2. Line 88: "as were individuals diagnosed ...", did it meant as well?

Reply: Thank you for pointing this out. We have rephrased the sentence into “Patients with unknown gender or missing data and individuals diagnosed with MVP (code 424.00) earlier than KCN were excluded.” in the revised manuscript on page 3, lines 96−98.

3. Line 89: Why chosed patients >12 years rather than other age?

Reply: Thank you for pointing this out. We chose KCN patients ≥12 years of age rather than other ages because KCN typically appears in adolescence. This has been mentioned in the Introduction section. We have added the statement in the revised manuscript on page 3, lines 98−99.

4. Line 100: "data regarding their comorbidities including hyperlipidaemia ...",  more studies should be cited to support the comorbidities selected for controls to determine whether it is appropriate.

Reply: Thank you for pointing this out. We have added the new reference studies to support the comorbidities selected for controls are appropriate and have cited the studies as new references 9 and 10 in the revised manuscript on page 3, line 112 and pages 11−12, lines 325−330.

  1. Andell, P.; Li, X.; Martinsson, A.; Andersson, C.; Stagmo, M.; Zoller, B.; Sundquist, K.; Smith, J.G. Epidemiology of valvular heart disease in a Swedish nationwide hospital-based register study. Heart 2017, 103, 1696-1703.
  2. Subki, A.H.; Bakhaidar, M.G.; Bakhaider, M.A.; Alkhowaiter, A.A.; Al-Harbi, R.S.; Almalki, M.A.; Alzahrani, K.A.; Fakeeh, M.M.; Subki, S.H.; Alhejily, W.A. Trends in mitral valve prolapse: a tertiary care center experience in Jeddah, Saudi Arabia. International journal of general medicine 2019, 12, 55-61.

Sincerely,

Ren-Long Jan

Department of Pediatrics, Chi Mei Medical Center, Liouying

201, Taikang, Taikang Village, Liouying District, Tainan 73657, Taiwan

Tel: +886-6-622-6999, ext. 77601

Fax: +886-6-283-2639, ext. 77610
